

# Using a genetic algorithm to find molecules with good docking scores

Casper Steinmann[1] and Jan H. Jensen[2]

[1] Department of Chemistry and Bioscience, Aalborg University, Aalborg, Denmark
[2] Department of Chemistry, University of Copenhagen, Copenhagen, Denmark

## ABSTRACT

A graph-based genetic algorithm (GA) is used to identify molecules (ligands) with high absolute docking scores as estimated by the Glide software package, starting from randomly chosen molecules from the ZINC database, for four different targets: *Bacillus subtilis* chorismate mutase (CM), human $\beta_2$-adrenergic G protein-coupled receptor ($\beta_2$AR), the DDR1 kinase domain (DDR1), and $\beta$-cyclodextrin (BCD). By the combined use of functional group filters and a score modifier based on a heuristic synthetic accessibility (SA) score our approach identifies between ca 500 and 6,000 structurally diverse molecules with scores better than known binders by screening a total of 400,000 molecules starting from 8,000 randomly selected molecules from the ZINC database. Screening 250,000 molecules from the ZINC database identifies significantly more molecules with better docking scores than known binders, with the exception of CM, where the conventional screening approach only identifies 60 compounds compared to 511 with GA+Filter+SA. In the case of $\beta_2$AR and DDR1, the GA+Filter+SA approach finds significantly more molecules with docking scores lower than $-9.0$ and $-10.0$. The GA+Filters+SA docking methodology is thus effective in generating a large and diverse set of synthetically accessible molecules with very good docking scores for a particular target. An early incarnation of the GA+Filter+SA approach was used to identify potential binders to the COVID-19 main protease and submitted to the early stages of the COVID Moonshot project, a crowd-sourced initiative to accelerate the development of a COVID antiviral.

## INTRODUCTION

Docking of molecules to protein targets is an important part of computer aided drug discovery (*Kitchen et al., 2004*). One use of molecular docking is high throughput virtual screening (HTVS) of libraries of known molecules. Recent studies have show that such HTVS of hundreds of millions (*Lyu et al., 2019*) or even billions of molecules (*Grebner et al., 2019*) are possible. However, such large numbers pale in comparison with the estimated $10^{60}$ small molecules that make up chemical space.

The only practical way to search this space is to use search algorithms to identify interesting sub-spaces of manageable sizes. Historically, most work in this area as it relates to drug discovery have used evolutionary search algorithms to address this problem and such methods have also been applied to docking. The use of evolutionary algorithms

Corresponding authors
Casper Steinmann, css@bio.aau.dk
Jan H. Jensen, jhjensen@chem.ku.dk

in drug discovery has been reviewed by *Devi, Sathya & Coumar (2015)*. Examples involving docking include work by *Pegg, Haresco & Kuntz (2001)*, *Nicolaou, Apostolakis & Pattichis (2009)*, and *Daeyaert & Deem (2016)* who all use genetic algorithms (GA) to optimise docking scores obtained by the DOCK (*Ewing et al., 2001*), Glamdock (*Tietze & Apostolakis, 2007*), Autodoc-Vina (*Trott & Olson, 2010*) programs, respectively. All three methods combine predefined molecular fragments in order to help ensure that the final molecules are synthetically accessible. Very recently, *Cofala et al. (2020)* and *Nigam et al. (2020)* presented SELFIES (*Krenn et al., 2020*)-based (mutations only) GA approaches for optimising docking scores. *Cofala et al. (2020)* optimised QuickVina 2 (*Alhossary et al., 2015*) docking scores for COVID-19 main protease ($M^{Pro}$). However, the several of the presented molecules in this study appear synthetically inaccessible. *Nigam et al. (2020)* optimised docking scores to 5-hydroxytryptamine receptor 1B and Cytochrome P450 2D6 by interpolating between a known binder to each target. Finally, *Cieplinski et al. (2020)* and *Boitreaud et al. (2020)* have used variational autoencoders (*Kusner, Paige & Hernández-Lobato, 2017*; *Gómez-Bombarelli et al., 2018*) to optimise SMINA (*Koes, Baumgartner & Camacho, 2013*) and Autodoc-Vina docking scores for several targets. *Cieplinski et al. (2020)* noted difficulties in finding good binders using this approach while *Boitreaud et al. (2020)* achieved some success.

In this paper we show that a non-fragment based GA (*Jensen, 2019*) can be used to find more synthetically accessible molecules with good Glide (*Friesner et al., 2004*; *Halgren et al., 2004*) docking scores compared to conventional HTVS of libraries. We note that our study does *not* address whether docking is useful for drug discovery.

## COMPUTATIONAL METHODOLOGY

A graph-based genetic algorithm (*Jensen, 2019*) (GA) is used to identify molecules (ligands) with high absolute docking scores as estimated by the Glide software package (*Friesner et al., 2004*; *Halgren et al., 2004*) using either the faster HTVS or the slower SP scoring methodology. While the scoring functions for HTVS and SP are the same, SP samples more intermediate conformations throughout the docking funnel, and also reduces the thoroughness of the final torsional refinement and sampling. For the COVID Moonshot project we also rescore some ligands using the XP scoring function, which has greater requirements for ligand–receptor shape complementarity and weeds out false positives that SP lets through. Five conformations of each molecule are generated using RDKit and minimized with the MMFF94 force field (*Halgren, 1996a*; *Halgren, 1996b*; *Halgren, 1996c*; *Halgren, 1996d*; *Halgren & Nachbar, 1996*). The lowest energy conformer is used for docking. The population size is 400 molecules, the mutation rate is 50% (meaning that a mutation operation is applied to 50% of the offspring-molecules), and the number of generations is 50. The maximum molecule size allowed is $30 \pm 5$ non-hydrogen atoms. Molecules are chosen for mating with a probability proportional to their scores (roulette selection) and the 400 best-scoring molecules are advanced to the next generation (elitism). The initial population is chosen randomly from a 250,000-molecule subset of the ZINC database (*Sterling & Irwin, 2015*) used in a previous study (*Jensen, 2019*).

As noted by *Gao & Coley (2020)* and *Brown et al. (2019)* generative models in general and GAs in particular often generate molecules with known chemically unstable bonds or molecules that are difficult to synthesise. We address this issue in three ways: we use Walters rd_filters code (following *Brown et al. (2019)*), a score modifier suggested by *Gao & Coley (2020)* based on a heuristic synthetic accessibility (SA) score (*Ertl & Schuffenhauer, 2009*), and a combination of the two. The rd_filers code contains several sets of SMARTS strings defining unstable bonds or groups. We use all the sets and eliminate any molecule with any of these moieties from the population. In the score modifier approach, the docking score is multiplied by a modified Gaussian function that ranges from 0 to 1 for high and low values of the SA score, respectively (a low SA score indicates a synthetically accessible molecule):

$$score := score \cdot e^{-\frac{1}{2}\left(\frac{\max(SA\ score, \mu) - \mu}{\sigma}\right)^2} \tag{1}$$

where $\mu = 2.230044$ and $\sigma = 0.6526308$ (*Gao & Coley, 2020*). We found that the heuristic SA score depends on the protonation state of acid/base groups and is lower (better) for the neutral protonation state, so we neutralise such groups before computing the SA score.

The synthetic accessibility of the molecules in the final populations are estimated using the Molecule.one software package (*Anonymous, 2020*). The calculations are submitted remotely to Molecule.one servers using a license generously provided by Molecule.one for this project. Just as for the heuristic SA score it is important to supply this algorithm with the neutralised forms of the molecules.

The docking targets are *Bacillus subtilis* chorismate mutase (CM), human $\beta_2$-adrenergic G protein-coupled receptor ($\beta_2$AR), the DDR1 kinase domain (DDR1), and $\beta$-cyclodextrin (BCD). For the three proteins we use the 2CHT (*Chook, Ke & Lipscomb, 1993*), 2RH1 (*Cherezov et al., 2007*), and 3ZOS (*Canning et al., 2014*) crystal structures from the Protein Data Bank (*Berman, 2000*), respectively. The proteins are prepared for docking with the Protein Preparation Wizard (*Sastry et al., 2013*) in the Maestro (*Schrödinger, LLC, 2019*) software by protonating all residues assuming pH 7 with PropKa (*Olsson et al., 2011*). All three protein structures contain co-crystallized ligands: a transition state analog (TSA) for CM, carazolol for $\beta_2$AR, and ponabtidin for DDR1. All water molecules in the crystal structures were removed before constructing a docking grid. For the three proteins, the position of each co-crystalized ligand is used as the centroid for the docking grid whereas for BCD we used the center of mass. A 20 Å buffer around the centroid was employed when constructing the docking grid. We re-dock these ligands to their respectively targets to get an idea of what docking score one would expect for known binders. We determined the protonation state and overall charge of each ligand ($-2$ for TSA, 0 for carazolol, and $+1$ for ponabtidin) by visual inspection of the crystal structure. In addition, we dock the known BCD-binder 6-(phenylamino)naphthalene-2-sulfonate (2,6-ANS) to BCD in the anionic form based on the typical pKa of the sulfonate group. The structures of the ligands are displayed in Table 1.

**Table 1** Docking scores in kcal/mol, heuristic synthetic accessibility scores, and Molecule.one scores for known binders

| Target | Molecule | | Charge | HTVS | SP | SA score | Molecule.one |
|---|---|---|---|---|---|---|---|
| CM | TSA | | −2 | −7.5 | −8.1 | 5.4 | 10.0 |
| $\beta_2$AR | Carazolol | | 0 | −6.8 | | 2.6 | 2.9 |
| DDR1 | Ponatinib | | +1 | −6.9 | | 2.9 | 2.3 |
| BCD | 2,6-ANS | | −1 | −4.4 | | 1.8 | 2.2 |

**Table 2** Columns 2–5 list the number of molecules (out of a total of about 8,000) with docking scores higher than known binders (Table 1) without any structural screening (GA), with the group-filters (+Filter), with a heuristic synthetic accessibility score (SA), and with both Filter and SA. The HTVS scoring methodology is used except for CM(SP) where the SP scoring methodology is used. Columns 6 and 7 list the corresponding number molecules with scores lower than −9.0 kcal/mol and −10.0 kcal/mol obtained with +Filter +SA. The last three columns list the corresponding number of molecules obtained by docking all the molecules from the 250 K ZINC subset.

| | GA | +Filter | +SA | +Filter + SA | < −9.0 | < −10.0 | ZINC | < −9.0 | < −10.0 |
|---|---|---|---|---|---|---|---|---|---|
| CM | 7,963 | 7,300 | 181 | 511 | 0 | 0 | 60 | 0 | 0 |
| CM(SP) | 4,638 | | | | | | | | |
| $\beta_2$AR | 7,994 | 7,879 | 2,493 | 2,125 | 164 | 10 | 16,262 | 86 | 1 |
| DDR1 | 7,940 | 7,239 | 2,469 | 2,119 | 378 | 38 | 11,713 | 199 | 8 |
| BCD | 8,000 | 7,947 | 6,214 | 6,218 | 0 | 0 | 152,209 | 0 | 0 |

## RESULTS AND DISCUSSION

We perform 20 different GA searches using the HTVS scoring function for each target. With a population size of 400 this generates up to 8,000 different potential binders for each target in the final populations (there are a few duplicates for some targets). Column 2 in Table 2 shows the number of molecules that have better (more negative) scores than known binders to the four targets (Table 1). Virtually all the molecules in the final populations have better scores than the known binders using the simpler HTVS scoring methodology. The average scores shown in Table 3 show that the scores are not only better, but considerably better, than for the known binders, except for CM where the decrease is more modest (0.8 kcal/mol compared to 3.1–4.7 kcal/mol). When using the more complex SP scoring function for CM the number of molecule with better scores drops to 4,638 compared to 7,963, indicating that any conclusions drawn below is likely to depend on the scoring function. However, given the computational expense of the SP scoring function and the relatively large number of docking simulations performed in this study we continue using the HTVS scoring function below.

While these results are encouraging, visual inspection of some of the best scoring molecules (first column of molecules in Fig. 1) show that they do not resemble drug-like

**Table 3** The average docking score and standard deviation (in kcal/mol) of all the molecules in the combined final populations of 20 GA searches, except for "ZINC" which lists the corresponding values for the 8,000 top scoring molecules obtained using the 250 K ZINC subset.

|  | GA | +Filter | +SA | +Filter + SA | ZINC |
|---|---|---|---|---|---|
| CM | $-8.3 \pm 0.4$ | $-8.7 \pm 0.7$ | $-5.8 \pm 1.0$ | $-5.8 \pm 1.3$ | $-6.1 \pm 0.4$ |
| CM(SP) | $-8.2 \pm 0.3$ | | | | |
| $\beta_2$AR | $-10.1 \pm 0.6$ | $-9.4 \pm 0.4$ | $-7.0 \pm 1.2$ | $-7.0 \pm 1.2$ | $-8.0 \pm 0.3$ |
| DDR1 | $-10.0 \pm 0.6$ | $-9.9 \pm 0.5$ | $-6.6 \pm 1.5$ | $-6.4 \pm 1.6$ | $-7.7 \pm 0.5$ |
| BCD | $-9.1 \pm 0.2$ | $-8.2 \pm 0.5$ | $-5.6 \pm 0.8$ | $-5.3 \pm 0.8$ | $-5.7 \pm 0.2$ |

molecules, indicating that they may be unstable and/or synthetically inaccessible. To quantify this property we compute a synthetic accessibility score using the Molecule.one retrosynthesis package which is based on machine learning algorithms and trained on a large number of reactions. Molecule.one returns a synthetic accessibility score ranging from 1 (very synthetically accessible) to 10 (not synthetically accessible) and the fraction of the 100 best-scoring molecules with a Molecule.one score below 10 is shown in Column 2 of Table 4. DDR1 is the only case for which a non-negligible fraction of molecules (26%) may be synthetically viable.

This problem has been observed before for generative models by, for example, *Gao & Coley (2020)*, *Brown et al. (2019)*, and *Renz et al. (2020)*. We address this issue in three ways: we use Walters rd_filters code (following *Brown et al. (2019)*), a score modifier based on a heuristic synthetic accessibility (SA) score (*Ertl & Schuffenhauer, 2009*) suggested by *Gao & Coley (2020)*, and a combination of the two. The results are shown in Table 2 and show that the use of filters has relatively little effect on the number of molecules with better scores than the known binders and their average docking score. Unfortunately, there is also a negligible effect on the fraction of molecules deemed synthetically accessible molecules by Molecules.one (Table 4). Inspection of the best scoring molecules for each target (second column of molecules in Fig. 1) reveals that while the filters successfully prevented unstable bonding patterns, they do not prevent other reactive moieties such as cyclopentadienes and cyclohexadiene-like motifs.

The use of the SA score modifier has a much bigger effect on the number of molecules with better scores than the known binders and their average docking score. The effect is most pronounced for CM where the number of good binders drops more than an order of magnitude to only 181 molecules, while the decrease is about 70% for both $\beta_2$AR and DDR1 and 23% for BCD. The most likely explanation is that ligands that bind well in the CM binding pocket tend to have high (bad) SA scores compared to the other targets. This is supported by the fact that the SA scores for the known binders TSA, carazolol, ponatinib, and 2,6-ANS (Table 1) are 5.4, 2.6, 2.9, and 1.8, respectively. The corresponding Molecule.one scores are 10.0, 2.9, 2.3, and 2.2, which indicates that the heuristic scores correlate well with the more sophisticated ML approach used by Molecule.one. With fewer molecules in the final population with high scores the average score necessarily increases (becomes less negative). The good news is that the fraction of molecules in the final population that Molecule.one deems synthetically accessible increases significantly to

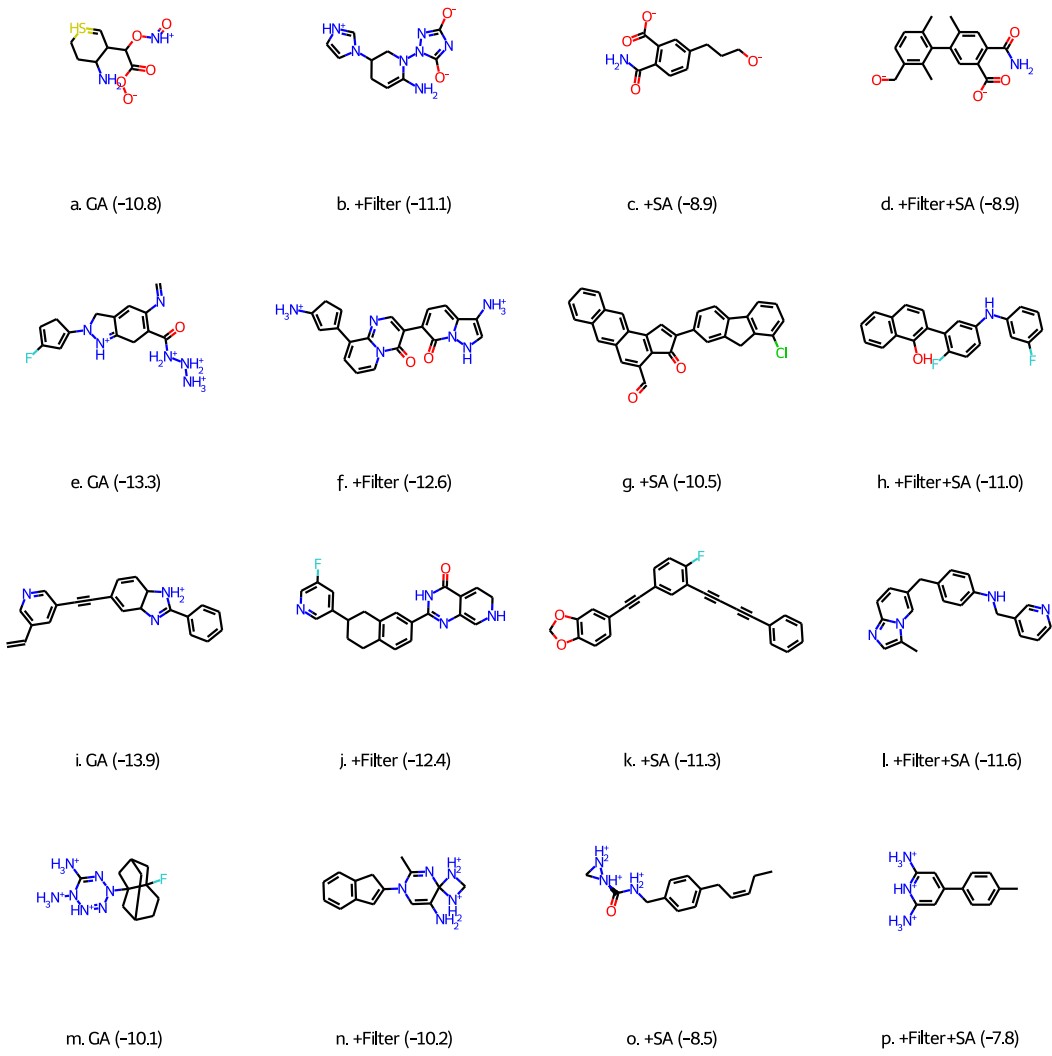

**Figure 1** **Best scoring molecules from the final population of GA-docking runs (see text for explanation) for the four different targets: CM (A–D), $\beta_2$AR (E–H), DDR1 (I–L), and BCD (M–P).** The scores (in kcal/mol) are shown in parentheses.

between 0.24 to 0.64. These fraction can be further increased to between 0.76 and 0.91, with only negligible effect on the number of good binders in the final population, by using the score modifier together with the filters. The increase in good binders (to 511) in the case of CM is most likely due to the stochastic nature of the GA searches.

A plot of the Molecule.one score vs the docking score (Fig. 2A) obtained using Filters+SA shows no correlation. Better scoring molecules are thus not necessarily harder to synthesise and the top scoring molecules for each target all have relatively low (good) synthetic accessibility scores. Furthermore, the fractions of synthetically accessible molecules

**Table 4 Fraction of the 100 top scoring molecules that are deemed synthetically accessible by Molecule.one.**

|  | GA | +Filter | +SA | +Filter + SA | ZINC |
|---|---|---|---|---|---|
| CM | 0.00 | 0.03 | 0.45 | 0.91 | 0.83 |
| $\beta_2$AR | 0.00 | 0.05 | 0.55 | 0.76 | 0.84 |
| DDR1 | 0.26 | 0.29 | 0.64 | 0.88 | 0.82 |
| BCD | 0.03 | 0.02 | 0.24 | 0.85 | 0.89 |

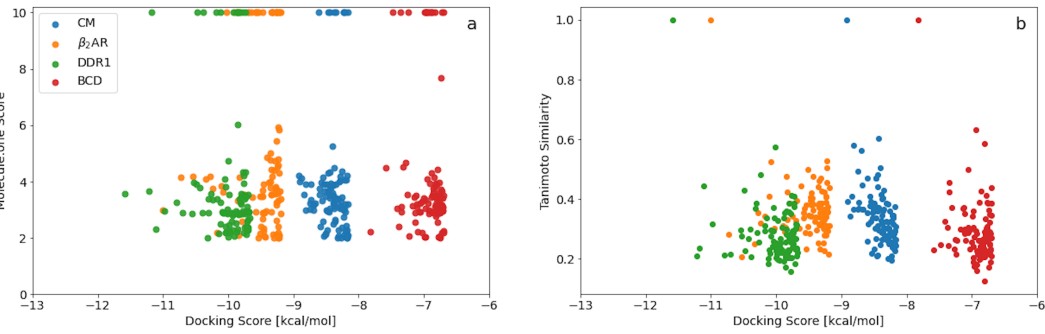

**Figure 2** (A) Molecule.one score versus docking score for the 100 best binders predicted for the four different targets using Filters and SA. (B) ECFP4 Tanimoto similarity to the best scoring molecule for each target vs docking score for the 100 best binders predicted for the four different targets using Filters and SA.

computed using the top 100 scoring molecules are thus expected to be representative of the corresponding fractions for the entire final population.

A similar plot of the ECFP4 Tanimoto similarity to the best scoring molecule for each target vs docking score (Fig. 2B) also shows no correlation. The minimum and maximum similarity to the best scoring molecules are in the range of about 0.2–0.6 and indicting a great deal of structural diversity among the 100 best scoring molecules for each target. The GA+Filters+SA docking methodology is thus effective in generating a large and diverse set of synthetically accessible molecules with high docking scores for a particular target.

Inspection of the top scoring molecules obtained with both filters and a score modifier (Fig. 1 (final column) and Fig. S2) reveal fairly ordinary looking organic molecules except that the charged states for CM and BCD (Figs. 1D and 1P) are not reasonable for a pH of 7. Future studies using this approach will need to correct this by, for example, including additional filters or adding a term to the score that penalizes large deviations from empirically estimated p$K_a$ values.

Finally, while the accuracy of the chosen docking methodology is not a focus of this paper we do note some encouraging signs for the HTVS scoring function in Glide. For example, all the top scoring molecules for CM (Fig. 1) are dianions just like the known binder TSA (Fig. 1) and the CM substrate chorismate. Only 327 out of the 250,000 molecules in the ZINC subset that the initial population is randomly drawn from are dianions so the dianion motif is most likely generated by the GA. Similarly, for the fluorene moiety seen in both carazolol and Fig. 1G as well as two ethylene linked aromatic moieties seen in ponatinib

and Figs. 1I and 1K, but with only 242 and 40 occurrences in the 250K ZINC subset, respectively. For BCD there is a clear preference for both lipophilic (adamantane-like in Fig. 1M) moieties favoring binding to the pocket as well as hydrophilic moieties which are also representative of the structures that bind favorably according to experiments.

## Comparison to conventional HTVS

While the GA+Filter+SA results are encouraging they do involve the docking of 400,000 compounds and the question whether equally good results can be obtained by simply screening a library of molecules of similar size. To answer this question we dock all 250,000 molecules in the ZINC subset from which we sample the initial population for the GA searches. This approach identifies significantly more molecules with better docking scores than known binders, with the exception of CM, where the conventional HTVS approach only identifies 60 compounds compared to 511 with GA+Filter+SA (Table 2). However, the known binders for the remaining targets have relatively low docking scores of less than $-7.0$ kcal/mol. In the case of $\beta_2$AR and DDR1 the GA+Filter+SA approach finds significantly more molecules with docking scores lower than $-9.0$ kcal/mol and $-10.0$ kcal/mol. Here, GA+Filter+SA finds 1.9 times as many molecules with a docking score lower than $-9.0$ kcal/mol by docking only 1.6 times as many molecules. Overall, the number of molecules deemed synthetically accessible by Molecule.one is about the same as for GA+Filter+SA: somewhat higher for $\beta_2$AR and BCD and somewhat lower for CM and DDR1 (Table 4, the top-10 scoring molecules for each target is shown in Fig. S3). The GA+Filter+SA therefore seems like a promising approach for finding molecules with very good docking scores compared to conventional HTVS of libraries.

## The COVID Moonshot project

The COVID Moonshot project (Chodera et al., 2020), a crowd-sourced initiative to accelerate the development of a COVID antiviral, was announced in mid-March 2020. The organizers of the project provided several crystal structures of the COVID-19 main protease ($M^{Pro}$) in complex with several small fragments (Douangamath et al., 2020) (Fig. S1) and invited the scientific community use this data to construct and submit potential $M^{Pro}$ inhibitors for further experimental verification. Though we were in a relatively early stage of this project we decided to build upon our methodology as it was then, to construct candidates for the second and third rounds of submissions.

For Round 2 we perform 20 GA searches with a population size of 400 using the HTVS+Filter+SA as above and the 6LU7 crystal structure of $M^{Pro}$ (Jin et al., 2020). However, at that early stage we sampled our initial population from the first 1,000 molecules of the 250K ZINC database subset. Also, we were not yet aware of the importance of neutralizing acid base groups before computing the SA score and checking synthetic accessibility, but this does not seem to be important for this target. Based on experience with the other targets we did increase the maximum molecule size to $50 \pm 5$ non-hydrogen atoms. All molecules in the final population are re-scored using the Glide XP scoring function (Friesner et al., 2006). The 128 molecules with a score better than or equal to $-7.0$ kcal/mol are selected and subjected to retrosynthetic analysis using the ASKCOS
software package (*Coley et al., 2019*) using the settings suggested by *Gao & Coley (2020)*. Molecules with less than 4 synthetic steps are selected and re-docked using the XP scoring methodology. The six molecules with XP scores better than −7.5 kcal/mol (Figs. 3A)–3F were then submitted to Round 2 in March 30th, 2020. The feedback on Twitter was that the molecules were rather small and more fragment-like than drug-like.

For Round 3 we use Glide SP rather than HTVS for the GA search, use Molecule.one in addition to ASKCOS to determine synthetic accessibility for molecules with XP scores better than −7.0 kcal/mol, and eliminate all molecules with for which both ASKCOS and Molecule.one fail to find a retrosynthetic route. The remaining 136 molecules are then re-docked using the XP scoring methodology and molecules are selected from among the top-scoring molecules. We selected four molecules for submission to Round 3 (Figs. 3G–3J) based on their score, diversity (also relative to our Round 2 submissions), and size and submitted them to Round 3 on April 2nd, 2020. (We also submitted four molecules selected based purely on their score, i.e., with synthetic accessibility considerations based on ASKCOS and Molecule.one, which are not discussed further).

Of the 10 submitted molecules, one was selected by the organizers (Fig. 3B) for synthesis and assay, but showed relatively low inhibition (10% average inhibition at 20 μM) and was not pursued further. Many of our submissions feature a urea linkage or amide linkage that are also present in many of the fragment binders identified by the COVID Moonshot organizers (Fig. S1). In fact one of our submissions (Fig. S3F) differs by only a few atoms from one of the fragments (Fig. S1L) as well as one the submissions selected for further study. Overall, these results are quite encouraging given that our submissions are generated starting from randomly selected molecules.

## CONCLUSION AND OUTLOOK

A graph-based genetic algorithm (*Jensen, 2019*) (GA) is used to identify molecules (ligands) with high absolute docking scores as estimated by the Glide software package (*Friesner et al., 2004*; *Halgren et al., 2004*), starting from randomly chosen molecules from the ZINC database. We perform 20 different GA searches using the HTVS scoring function each for four different targets: *Bacillus subtilis* chorismate mutase (CM), human -adrenergic G protein-coupled receptor ($\beta_2$AR), the DDR1 kinase domain (DDR1), and $\beta$-cyclodextrin (BCD). With a population size of 400 this approach generates up to 8,000 different potential binders for each target, almost all of which have a better docking score than known binders (Figs. 1 and 2). However, many of these molecules do not resemble drug-like molecules (Fig. 1 and Fig. S2) and virtually none of the top-100 scoring molecules are deemed synthetically accessible by the retrosynthetic software package Molecule.one (*Molecule.one, 2020*) (Table 4).

Following suggestions by *Brown et al. (2019)* and *Gao & Coley (2020)* we show that the synthetic accessibility can be increased significantly by the combined use of Walters rd_filters code and a score modifier based on a heuristic synthetic accessibility (SA) score (*Ertl & Schuffenhauer, 2009*) (GA+Filter+SA). However, this also leads to a drop in the number of molecules with scores better than known binders of between 22% (BCD) and

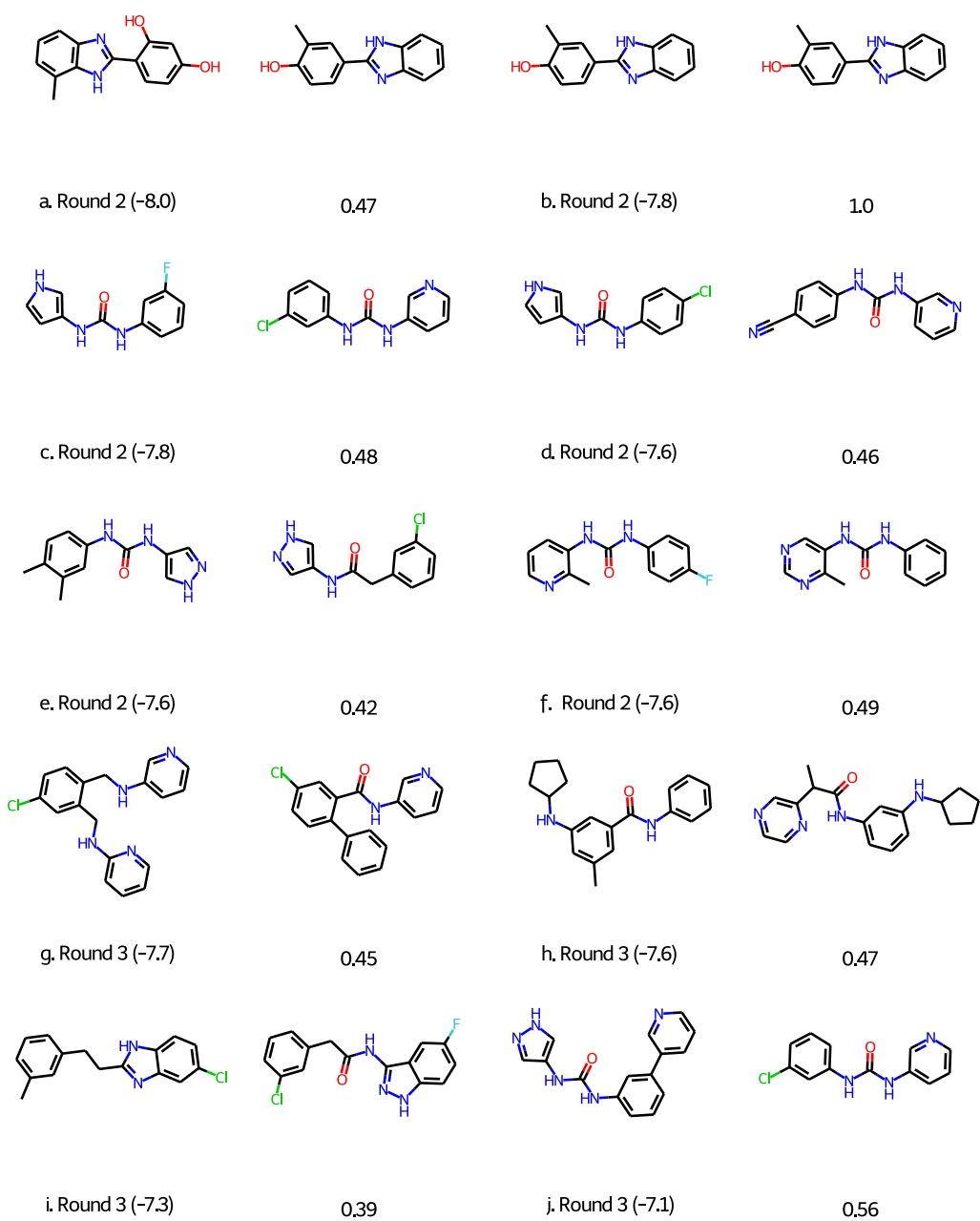

**Figure 3 Molecules submitted to the second and third round of the COVID moonshot project (with the corresponding XP docking score in parenthesis).** Next to each submitted molecule is a molecule that was selected for further study by the project organizers that most closely matches our submissions (with the corresponding ECFP4 Tanimoto similarity below). All docking scores in kcal/mol.

95% (CM). The GA+Filter+SA approach thus identifies between roughly 500 and 6000 structurally diverse (Table 2 and Fig. S2) molecules with scores better than known binders by screening a total of 400,000 molecules starting from 8000 randomly selected molecules from the ZINC database. However, screening 250,000 molecules from the ZINC database

identifies significantly more molecules with better docking scores than known binders, with the exception of CM, where the conventional HTVS approach only identifies 60 compounds compared to 511 with GA+Filter+SA (Table 2). The known binders for the remaining targets have relatively low docking scores of less than −7.0 kcal/mol. In the case of $\beta_2$ AR and DDR1 the GA+Filter+SA approach finds significantly more molecules with docking scores lower than −9.0 kcal/mol and −10.0 kcal/mol. The GA+Filters+SA docking methodology is thus effective in generating a large and diverse set of synthetically accessible molecules with very good docking scores for a particular target. However, for targets such as CM that predominantly binds charged ligands this approach will need to correct for unphysical protonation states by, for example, including additional filters or adding a term to the score that penalizes large deviations from empirically estimated p$K_a$ values.

An early incarnation of the GA+Filter+SA approach was used to identify potential binders to the COVID-19 main protease (M$^{Pro}$) and submitted to the early stages of the COVID Moonshot project (*Chodera et al., 2020*), a crowd-sourced initiative to accelerate the development of a COVID antiviral. Of the 10 submitted molecules, one was selected by the COVID Moonshot organizers (Fig. 3B) for synthesis and assay, but showed relatively low inhibition (10% average inhibition at 20 μM) and was not pursued further. Many of our submissions feature a urea linkage or amide linkage that are also present in many of the fragment binders identified by the COVID Moonshot organizers (Fig. S1). In fact one of our submissions (Fig. 3F) differs by only a few atoms from one of the fragments (Fig. S1L) as well as one the submissions selected for further study. Overall, these results are quite encouraging given that our submissions are generated starting from randomly selected molecules.

As pointed out by *Cieplinski et al. (2020)* docking scores may also be used as a challenging test for generative models that "reflect the complexity of real discovery problems" (*Coley, Eyke & Jensen, 2020*). Our study suggests that finding synthetically accessible molecules with good docking scores for CM presents an especially challenging objective function, and more so if the SP docking score is used. However, a benchmark based on a commercial software package such as Glide is not ideal and it remains to be seen whether this target is equally challenging using open source docking software such as SMINA (*Koes, Baumgartner & Camacho, 2013*).

### Funding
The authors received no funding for this work.

### Competing Interests
Jan H. Jensen is an Academic Editor for PeerJ.

## Author Contributions

- Casper Steinmann conceived and designed the experiments, performed the experiments, analyzed the data, performed the computation work, prepared figures and/or tables, authored or reviewed drafts of the paper, and approved the final draft.
- Jan H. Jensen conceived and designed the experiments, analyzed the data, performed the computation work, prepared figures and/or tables, authored or reviewed drafts of the paper, and approved the final draft.

## Data Availability

The code used in this study is available at GitHub: https://github.com/cstein/GB-GA/tree/feature-glide_docking.

SMILES strings, docking scores, and Molecule.one scores are available at GitHub: https://github.com/cstein/GB-GA_docking_supporting_information.

## Supplemental Information

Supplemental information for this article can be found online at http://dx.doi.org/10.7717/peerj-pchem.18#supplemental-information.

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
