# Peer review of "Using a genetic algorithm to find molecules with good docking scores"

_PeerJ Physical Chemistry, doi:10.7717/peerj-pchem.18_

## Round 0.1 · original submission · Minor Revisions

Both reviewers asked for clarifications on several methodological details. Please revise the manuscript accordingly.

·

Basic reporting

Steinmann et al. demonstrate the use of a genetic algorithm to design new structures in drug discovery using docking. Compared to traditional approaches such as high throughput virtual screening, the authors demonstrate superior performance in light of synthetic feasibility of generated molecules. I recommend acceptance of this article after incorporation of the following minor edits:

- In one of the experiments, the authors report the following fitness function “In the score modifier approach, the docking score is multiplied by a modified Gaussian function that ranges from 0 to 1 for high and low values of the SA score”. Here, it is my opinion that explicitly stating the fitness functions (using equations) for all experiments will help a reader. I found it challenging to understand the equation for this particular example.

- “the mutation rate is 50%”: Can the authors please clarify how the mutation rate is determined?. For example, for a given molecule, are 50% of the atoms and bond modified?. In my opinion, this will help readers that have not read the GB-GA paper. Additionally, was this 50% mutation rate also used for crossovers?.

- “While these results are encouraging visual inspection of some of the best scoring molecules (first column of molecules in Figure 1) show that they do not resemble drug-like molecules, indicating that they may be unstable and/or synthetically inaccessible.” For better readability, please add a comma after “encouraging” (or break the sentence into a few parts).

- “ and the fraction of the 100 best-scoring molecules with a Molecule.one score below 10 is shown in Column 2 of Table 4” In the previous part of this sentence, the authors state that Molecule.one returns a score between 1 and 10. As such, re-stating that “100 best-scoring molecules with a Molecule.one score below 10” might be redundant. Maybe the authors wanted to convey something else?

- “ The only case for which a non-negligible fraction of molecules may be synthetically viable is DDR1 with 26% while for the rest that fraction is very close to 0%.”. Please consider rewriting this sentence. It is hard to read.

- “A similar plot of the Tanimoto similarity to the best scoring molecule for each target vs docking score (Figure 2b) also shows no correlation”. The authors should explicitly state which fingerprint was utilized for calculating the similarity.

AkshatKumar (Akshat) Nigam

Experimental design

No comment

Validity of the findings

No comment

Additional comments

No comment

·

Basic reporting

In the manuscript ‘Using a genetic algorithm to find molecules with good docking scores’ the authors Steinmann and Jensen present a graph-based genetic algorithm (GA) based approach to identify highly scoring molecules) from the ZINC database for different receptor targets. The approach combines a functional group filter with a docking score modifier (based on a heuristic synthetic accessibility score, SA) (GA+Filter+SA approach). The authors compare their results to standard screening of large molecular datasets.
The paper tackles the challenge of efficiency for virtual screening techniques of millions of docking compounds with the target to select lead structures that are possible to synthesize chemically. The introduction refers to several recently developed and applied methodologies, mostly based on fragment-based techniques in combination with GA search algorithms. The authors highlight, that their approach is different as it is not based on a fragment-based approach and focuses on better selection of synthetic assessable molecules.
The basic reporting in the manuscript is clear and coherent.

Experimental design

The design of the study and the computational methodology of the paper is sound. It could be considered if a flow diagram for the methodology might help the reader in the methodology section. Some minor questions on the details might be interesting to add to the methodology section as listed below:
Is there a specific reason that Glide has been chosen as docking software?
It might be valuable for the reader to briefly comment on the difference between Glide’s HTVS and SP (and XP) scoring.
Why have five conformations been generated for each molecule at the start and how does this compare to e.g. a standard combination of Glide SP + Glide XP conformational search and optimisation approach? (also in terms of computational costs). Also in relation to the different number of flexibly rotating bonds in different molecules, could this fixed number lead to a bias in conformer selection?
“the docking score is multiplied by a modified Gaussian function that ranges from 0 to 1 for high and low values of the SA score”: Would it be possible to inform very briefly on the type of modification?
On the receptors: Do the receptors contain any “specialities” like metal ions or water molecules in the binding pockets and have they been treated (or removed)? Has a docking grid been defined and what was its centroid and size?

Validity of the findings

The findings are valid and relevant. Independent from the fact that a zoo of virtual screening methodologies is published in literature a focus on including synthetic accessibility better into high throughput methods is valuable.

Additional comments

As the authors might know, this referee is a strong supporter of open and FAIR data sharing principles. Thus, I appreciate that github links have been included in the supporting information. However, the links seem to be broken and might need corrections. (Is the interface with graph-based GA algorithm for this application also included there?)
The link between the new approach and the Covid Moonshot project could be established (on page 7) a bit better. It appears that “An early incarnation of the GA+Filter+SA approach” has been used for this part of the study and it might be good to better understand why this then has been added to this manuscript and if the developed approach would result in different outcomes.
Other minor comments:
Please add units to values in Tables
Figure 1 Caption: Replace “Highest scoring” with “best scoring”?
Page 6 (Comparison to…): “equally good results can obtained” … “can be obtained”

---

## Round 0.2 · accepted · Accept

I'm happy to accept the revised manuscript for publication.